# C-Reactive Protein and Serum Albumin Ratio: A Feasible Prognostic Marker in Hospitalized Patients with COVID-19

**DOI:** 10.3390/biomedicines10061393

**Published:** 2022-06-13

**Authors:** Vicente Giner-Galvañ, Francisco José Pomares-Gómez, José Antonio Quesada, Manuel Rubio-Rivas, Javier Tejada-Montes, Jesús Baltasar-Corral, María Luisa Taboada-Martínez, Blanca Sánchez-Mesa, Francisco Arnalich-Fernández, Esther Del Corral-Beamonte, Almudena López-Sampalo, Paula María Pesqueira-Fontán, Mar Fernández-Garcés, Ricardo Gómez-Huelgas, José Manuel Ramos-Rincón

**Affiliations:** 1Department of Internal Medicine, Hospital Clínico Universitario San Juan de Alicante, 03550 Alicante, Spain; 2Fundación para el Fomento de la Investigación Sanitaria y Biomédica (FISABIO), 46020 Valencia, Spain; pomares_fra@gva.es; 3Departamento de Medicina Clínica, Medicine School, University Miguel Hernández, 03550 Alicante, Spain; jquesada@umh.es (J.A.Q.); jose.ramosr@umh.es (J.M.R.-R.); 4Department of Endocrinology, Hospital Clínico Universitario San Juan de Alicante, 03550 Alicante, Spain; 5Department of Internal Medicine, Bellvitge University Hospital, 08097 L’Hospitalet de Llobregat, Spain; mrubio@bellvitgehospital.cat; 6Department of Internal Medicine, Hospital Universitario 12 de Octubre, 28041 Madrid, Spain; javiertejadamontes@gmail.com; 7Department of Internal Medicine, Hospital Gregorio Marañón, 28007 Madrid, Spain; jesus.baltasar@salud.nadrid.org; 8Department of Internal Medicine, Hospital Universitario de Cabueñes, 33394 Gijón, Spain; mlmartinez72@gmail.com; 9Department of Internal Medicine, Hospital Costa del Sol, 20603 Marbella, Spain; sanchezmesa.blanca@gmail.com; 10Department of Internal Medicine, Hospital Universitario La Paz, 28046 Cantoblanco, Spain; arnalich@salud.madrid.org; 11Department of Internal Medicine, Hospital Royo Villanova, 50015 Zaragoza, Spain; esdcorral@gmail.com; 12Department of Internal Medicine, Regional University Hospital of Málaga, 29010 Málaga, Spain; almu_540@hotmail.com (A.L.-S.); ricardogomezhuelgas@hotmail.com (R.G.-H.); 13Biomedical Research Institute of Málaga (IBIMA), University of Málaga (UMA), 29590 Málaga, Spain; 14Department of Internal Medicine, Complejo Hospitalario Universitario de Santiago, 15706 Santiago de Compostela, Spain; paulapesqueira@hotmail.com; 15Department of Internal Medicine, Doctor Peset University Hospital, 46017 Valencia, Spain; mar.fernandez.7823@gmail.com

**Keywords:** COVID-19, serum albumin, C-reactive protein, prognosis, syndemic, comorbidity

## Abstract

(1) Background: C-reactive protein (CRP) and albumin are inflammatory markers. We analyzed the prognostic capacity of serum albumin (SA) and CRP for an outcome comprising mortality, length of stay, ICU admission, and non-invasive mechanical ventilation in hospitalized COVID-19 patients. (2) Methods: We conducted a retrospective cohort study based on the Spanish national SEMI-COVID-19 Registry. Two multivariate logistic models were adjusted for SA, CRP, and their combination. Training and testing samples were used to validate the models. (3) Results: The outcome was present in 41.1% of the 3471 participants, who had lower SA (mean [SD], 3.5 [0.6] g/dL vs. 3.8 [0.5] g/dL; *p* < 0.001) and higher CRP (108.9 [96.5] mg/L vs. 70.6 [70.3] mg/L; *p* < 0.001). In the adjusted multivariate model, both were associated with poorer evolution: SA, OR 0.674 (95% CI, 0.551–0.826; *p* < 0.001); CRP, OR 1.002 (95% CI, 1.001–1.004; *p* = 0.003). The CRP/SA model had a similar predictive capacity (honest AUC, 0.8135 [0.7865–0.8405]), with a continuously increasing risk and cutoff value of 25 showing the highest predictive capacity (OR, 1.470; 95% CI, 1.188–1.819; *p* < 0.001). (4) Conclusions: SA and CRP are good independent predictors of patients hospitalized with COVID-19. For the CRP/SA ratio value, 25 is the cutoff for poor clinical course.

## 1. Introduction

COVID-19 is an acute viral disease that is characterized by a systemic hyperinflammatory state with marked pulmonary damage as the main prognostic factor [1,2]. The high interindividual heterogeneity in the clinical course is also noteworthy, with about 20% to 28% of patients requiring hospitalization [3,4] and about 5% requiring admission to the intensive care unit (ICU) [5]. Although many retrospective observational studies have shown comorbidity and age to be the main prognostic markers, there are currently no markers to predict the course of the infection in an individual patient [6,7]. This lack was also observed in studies applying big data methodology [8,9].

The serum albumin (SA) concentration is affected during acute diseases, especially in relation to systemic inflammation, in which levels are usually low. The mechanisms underlying this observation are different. The first involves malnutrition associated with the hypercatabolic state in acute disease. A second cause is that, although hepatic synthesis is increased in inflammatory processes, circulating albumin is low due to increased capillary permeability, with the consequent transmembrane loss of the molecule to the interstitial space. A third inflammation-related mechanism causing hypoalbuminemia is increased hepatic and predominant intracellular degradation, with a consequent reduced albumin half-life leading to a real decrease in body albumin content. Therefore, hypoalbuminemia can be considered an inflammatory marker [10,11] and has also been considered a global marker of general functional status, connecting inflammation, nutrition, and individual general status, and thus indicating the individual’s capacity to fight against infection [12]. Such an approach is important when addressing COVID-19, not only in terms of a pandemic, but also as a syndemic situation [13].

The initial population-based general observational retrospective studies showed lower SA levels to be related to poorer prognosis in patients with COVID-19 [12,14,15,16]. These initial studies included SA among different analytical biomarkers, with a poor quality in their conclusions, due to high methodological heterogeneity, although aiming at the potential prognostic value of SA. It also has to be considered that these initial studies were made in Chinese populations, with an evident limitation considering the potential generalization of the findings. Similar results were obtained in posterior specific studies not only in Chinese [17,18] but also in Caucasian populations [19,20,21,22,23,24,25]. The main limitation of these studies was the usual small sample size, as shown in recent meta-analysis, that confirmed the utility of SA as a prognostic marker in hospitalized acute COVID-19, although again indicating a very high heterogeneity of the analyzed literature, as well as small samples [26,27]. Elsewhere, authors have tried to improve the prognostic capacity of SA by analyzing ratios including other analytical parameters alone [28,29,30,31,32,33,34,35]. As a whole, these studies indicate an increased prognostic value than SA alone; however, samples have generally been small, thus limiting the strength of the findings reported in studies [36].

The objective of the present study was to analyze the predictive capacity of SA, CRP, and the combination of both variables as prognostic markers in hospitalized COVID-19 patients. Our hypothesis was that the combination of both variables could increase the predictive capacity of each variable alone.

## 2. Materials and Methods

### 2.1. Type of Study

The present study was a retrospective cohort study of patients from the Spanish SEMI-COVID-19 Registry, an ongoing, nationwide, multicenter, observational, retrospective cohort registry [37]. More than 150 hospitals from the 17 regions of Spain participate in the registry, thus ensuring a representative national sample. All consecutive hospitalized patients with confirmed SARS-CoV-2 infection who were discharged or died during hospitalization were eligible for inclusion. The inclusion criteria were age ≥18 years and first admission to the hospital due to SARS-CoV-2 infection confirmed microbiologically by real-time reverse transcription polymerase chain reaction (RT-PCR) testing of a nasopharyngeal or sputum sample or through a positive result on a serological test with a clinically compatible presentation, according to the recommendations of the World Health Organization (WHO) [38]. The exclusion criteria were subsequent admissions of the same patient and denial or withdrawal of informed consent. Patients were admitted and treated at the discretion of the attending physicians based on their clinical judgment, local protocols, and the updated recommendations of the Spanish Ministry of Health. Personal-data processing complied with applicable European Union and Spanish laws on biomedical research and personal data protection. The Registry was approved by the Provincial Research Ethics Committee of Málaga (Spain) according to the recommendation of the Spanish Agency of Medicines and Medical Devices (AEMPS (Spanish abbreviation)). All patients gave their informed consent. When there were biosafety concerns and/or when the patient had already been discharged, verbal informed consent was requested and noted on the medical record. The conduct and reporting of the study was carried out pursuant to the STROBE statement guidelines [39].

### 2.2. Procedures

An online electronic data capture system (DCS) was developed for the SEMI-COVID-19 Registry. After training, at least one physician from the internal medicine department in each participating hospital was responsible for acquiring and entering the requested data into the DCS. This work was carried out on a voluntary basis, and physicians received no remuneration for it. In order to ensure the quality of data collection, a database manager was appointed, and data verification procedures were designed. The database was monitored by a scientific steering committee and an independent external agency. Similarly, data were analyzed and logistics were coordinated by independent external agencies. Alphanumeric sequences of characters were used as identification codes to pseudonymize dissociated patient identifiable data; as such, the DCS did not contain any direct patient identifiers. The database platform is hosted on a secure server, and all information is fully encrypted through a valid TLS certificate.

A total of 321 variables were retrospectively collected in the registry under different headings: (1) inclusion criteria, (2) epidemiological data, (3) RT-PCR and SARS-CoV-2 serology data, (4) personal medical and medication history, (5) symptoms and physical examination findings at admission, (6) laboratory values (blood gases, metabolic panel, complete blood count, coagulation) and diagnostic imaging tests, (7) additional data at seven days after admission to hospital or at admission to the ICU, (8) pharmacological treatment during hospitalization and need for ventilator support, (9) complications during hospitalization, and (10) progress after discharge and/or 30 days after diagnosis. A full list of the variables gathered can be found in the source paper [37].

A total of 5007 consecutive patients were recruited from March to June 2020. Patients without registered CRP and/or SA values at admission were excluded. Improbable extreme values were removed for each quantitative variable. Variables with considerable missing data and those of in-hospital complications and COVID-19-specific pharmacological treatments were also eliminated for the actual study. The final sample comprised 3471 patients (69.3% of all the patients in the registry at the time of the analysis), and 88 covariates were analyzed (Appendix B).

### 2.3. Exposure and Outcome

The two baseline exposure variables were SA (g/dL) and CRP (mg/L). The final outcome was a composite of in-hospital all-cause mortality, ICU admission, need for non-invasive mechanical ventilation (NIMV), and length of stay ≥14 days (the 75th percentile value of this variable). Follow-up was from the day of admission until in-hospital death or hospital discharge.

### 2.4. Statistical Analysis

A descriptive analysis was carried out by calculating the frequencies for qualitative variables and the mean and standard deviation for quantitative variables. The factors associated with the presence of the main outcome were analyzed by using contingency tables. The mean values of qualitative variables were compared by using the chi-square test, and those of quantitative variables by using the *t*-test. Multivariate logistic models were fitted to estimate the magnitude of the associations with the final outcome, and the odds ratios (ORs) were estimated with their 95% confidence intervals (CIs). Stepwise variable selection was carried out based on the Akaike information criterion. Two multivariate logistic models were adjusted with the same covariates to estimate the predictive capacity of SA and CRP for the main endpoint, as individual variables, as a composite form (SA + CRP), and as a ratio (CRP/SA). The optimal predictive cutoff point for CRP/SA was calculated. After a final validation of the model obtained, the total sample was randomly divided into a training group (70%) to build the model and a testing group (30%) to validate it. Validation consisted of using the model adjusted in the training group with the testing group data and calculating the probability of occurrence of the final outcome. With this probability and the final outcome observed in testing, the receiver operating characteristic (ROC) curve was calculated as an honest predictive indicator of the model, along with its 95% CI. The statistical analysis was carried out by using R, version 4.0.0, R Core Team, Vienna, Austria.

## 3. Results

### 3.1. Characteristics of the Study Population

As illustrated in Table 1, 57.6% of the 3471 patients were male, with a mean (range) age of 66 (20–102) years. Significant functional dependence (Barthel score) was recorded in 14.1% of the participants, and comorbidity (age-adjusted Charlson scale) was present in 57.6%, with high individual variability (mean 3.4 points, range [0.0–16.0] points). The most frequent previously used pharmacological treatments were statins (32.6%), angiotensin receptor blockers (18.8%), angiotensin-converting enzyme inhibitors (16.3%), and metformin (13.1%).

### 3.2. COVID-19-Related Characteristics

Almost 90% of the patients were Caucasian, and most infections (85.9%) were acquired in the community. Patients were admitted a mean (SD) of 6.9 (4.4) days after onset of COVID-19-related symptoms, with 11.6 (9.3) days of hospital stay; for 26.9% of the participants, the hospital stay was longer than 14 days. The most common symptoms were dry cough (59.7%) and dyspnea (58.2). At admission, most patients (64.5%) had an axillary temperature higher than 38.0 °C, with a mean baseline O_2_ saturation of 93.3% (5.5%) and crackles as the commonest finding on chest auscultation (52.5%). Plain chest X-ray examinations were normal in approximately half of the cases, with bilateral interstitial infiltrate as the commonest pathological finding (56.3%).

### 3.3. Outcomes

The final composite outcome was reached by 1425 participants (41.1%). NIMV was necessary in 5.8%, and 9.2% were admitted to the ICU. As previously stated, the length of stay was analyzed by stratifying it as more or fewer than 14 days, with the length of stay being longer than 14 days in 26.9% of cases. In-hospital, all-cause mortality was reported in 17.6% of cases.

### 3.4. Univariate Analysis

As expressed in Table 1 and Table 2, patients who developed the final outcome were older (72.0 [14.7] years vs. 62.1 [15.6] years; *p* < 0.001), with a higher proportion of males (61.5% vs. 54.9%; *p* < 0.001) and a predominance of Caucasians in both groups. Most patients in both groups acquired the infection in the community (80.8% vs. 89.5% for patients who reached and did not reach the final outcome, respectively). When baseline comorbidities were analyzed, patients for whom the final outcome was recorded were more likely to have chronic hypertension (60.7% vs. 41.7%; *p* < 0.001), diabetes mellitus (24.2 vs. 15.2; *p* < 0.001), atrial fibrillation (15.4% vs. 6.9%; *p* = 0.001), neurodegenerative diseases (12.0% vs. 4.8%; *p* < 0.001), chronic heart failure (10.2% vs. 3.9%; *p* < 0.001), chronic kidney disease (10.1% vs. 3.9%; *p* < 0.001), and significant functional dependence (Barthel scale) (22.5% vs. 8.3% patients; *p* < 0.001). Before admission, patients who reached the final outcome were more frequently receiving systemic corticosteroids (7.1% vs. 2.7%; *p* < 0.001), angiotensin receptor blockers (22.9% vs. 15.9%; *p* < 0.001), statins (38.0% vs. 28.8%; *p* < 0.001), aspirin (17.7% vs. 10.9%), and different anticoagulants.

When considering the COVID-19-related variables, we see that the participants for whom the final outcome was recorded had significantly higher values for CRP (108.9 [96.5] mg/L vs. 70.6 [70.3] mg/L; *p* < 0.001) and lower values for SA (3.5 [0.6] g/dL vs. 3.8 [0.5] g/dL; *p* < 0.001). Hypoalbuminemia, defined as SA < 3.5 g/dL, was recorded in 42.4% of the patients who reached the final outcome, and in 21.2% of those who did not (*p* < 0.001). A significant difference was detected between groups when considering the CRP/SA ratio (31.9 [29.2] for patients who reached the final outcome vs. 19.2 [20.2] for those who did not; *p* < 0.0001).

### 3.5. Multivariate Analysis

In order to fit a multivariate logistic model, about 70% of the initially included participants (n = 2485) were considered as the training group. A total of 88 potential explanatory variables were included in the model, which covered SA and CRP values; demographic characteristics; baseline comorbidities; and clinical, radiological, and other analytical COVID-19-associated variables.

As illustrated in Table 3, for the additive model (SA + CRP), both SA (OR, 0.674; 95% CI, 0.551–0.826; *p* < 0.001) and CRP values (OR, 1.002; 95% CI, 1.001–1.004; *p* = 0.003) persisted as explanatory variables after adjustment. The area under the ROC curve (AUC) for the estimated model was 0.8223 (95% CI, 0.8058–0.8388; *p* < 0.001), which was close to the values obtained when the model was applied to the random testing sample containing about 39% of the initial patients (n = 985) (AUC, 0.8176; 95% CI, 0.7908–0.8444; *p* < 0.001), thus validating the model. The model enabled us to estimate that, for each increase of 1 g/dL in the SA value, the adjusted risk for the final composite endpoint decreases by 32.6%. For CRP, each 1 mg/L increase was related to a 0.2% increase in risk.

A sensitivity analysis based on the optimal multivariate model was performed to evaluate the predictive capacity of SA and CRP for the main outcome. The model was adjusted for SA without CRP (AUC, 0.8214), CRP without SA (AUC, 0.8198), and the CRP/SA ratio (AUC, 0.8196). All four models yielded a similar AUC (about 0.82), although the highest area was obtained with the initial multivariate model applied to the additive form SA + CRP (AUC, 0.8224) (Figure 1).

In order to facilitate clinical applicability of the previous findings, a cutoff value of a simple and compelling parameter, the CRP/SA ratio, was analyzed as a prognostic factor. The associated OR and AUC were estimated for different cutoff values of this variable (Figure 2). The estimations revealed that cutoff values between 10 and 83 were associated with a significantly increased risk, with 77 being the value associated with the highest risk (OR, 2.257; 95% CI, 1.290–3.948; *p* = 0.004; AUC, 0.8182). Our analysis of the model using the likelihood ratio test revealed an adequate goodness of fit for values between 10 and 90. When the estimated AUC was considered, the optimal predictive capacity was observed with a cutoff value of 25 (OR, 1.470; 95% CI, 1.188–1.819; *p* < 0.001).

## 4. Discussion

The main finding in the present study was that hospitalized COVID-19 patients have a worse prognosis with lower SA values and higher CRP values at admission, irrespective of other confounding variables. The combination of both parameters enhances the predictive capacity of each variable alone, with a cutoff value of about 25 for the variable CRP/SA ratio as an integrative prognostic indicator, although a significantly increased direct risk is continuously observed for values of about 10 or higher.

The COVID-19 pandemic has underscored the need to identify patients who will progress poorly. In fact, identification of individuals with a worse prognosis can also help to organize healthcare resources [40,41,42]. Since the early stages of the pandemic, a series of clinical and analytical prognostic variables were identified from retrospective descriptive studies with the objective of obtaining a good prognostic marker. Comorbidity and age were generally considered the main prognostic factors [43,44]. Once prognostic variables were identified, multifactorial models were published to enhance their predictive capacity separately [45,46]. Models based on artificial intelligence have also been developed [8,9]. Two of the most widely used in clinical practice multifactorial models are the CALL Score [45] and the ISARIC-4C one [46]. The CALL score includes preexisting comorbidities, age over 60 years, lymphocyte count, and lactate dehydrogenase as markers of disease progression [45]. ISARIC-4C integrates 11 demographic, analytical, and clinical variables to predict future progress at diagnosis [46]. The complexity arising from the need to include multiple variables and the need to calculate the related score limits application in daily clinical practice. Furthermore, since these prognostic scores have been validated for specific populations with specific sociodemographic and structural healthcare conditions and resources, application needs to be validated for the population under study [47]. In the present study, as reported elsewhere, patients with a poorer clinical course were older with more comorbid conditions, as expressed by a higher age-adjusted Charlson score. As a difference of the present study with previous studies, while fully validated for measuring and defining comorbidity, the Charlson score has received little attention in publications on COVID-19 [48].

COVID-19 is essentially an inflammatory process, with CRP and other inflammatory parameters considered prognostic markers [1,2,49], including mainly CRP and also SA, as demonstrated in the most recent meta-analysis, although with a low quality of the evidence, mainly due to the fact that studies are all retrospective and with a usual small sample size [35]. SA is easily identified as a nutritional marker, yet the fact that it is also influenced by systemic inflammation means that it can also be considered an inflammatory marker [10,11]. As a consequence, low levels of albumin indicate a weakened general status and diminished host immune response, which is a well-known factor associated with comorbidity [12]. In accordance with this observation, an increasing number of studies that have specifically examined the potential predictive capacity of SA in COVID-19 have shown this parameter to be a good course indicator in terms of clinical deterioration, need for admission to the ICU, and mortality in patients with low SA values [12,14,15,16,17,18,19,20,21,22,23,24,25].

Since the first main retrospective Chinese publications analyzing potential prognostic factors, all publications have pointed toward SA as a good predictor in COVID-19. In accordance with this, a retrospective Chinese study of 78 patients found that patients whose clinical condition improved during hospitalization had higher albumin values (4.1 [4.5] g/dL) than those whose disease progressed (3.6 [6.6] g/dL) [18]. With respect to hospitalized patients, Li et al. [12] observed a significantly lower SA in critically ill patients (3.4 [3.1–3.9] g/dL) and severely ill patients (3.7 [3.4–4.2] g/dL) than in those who were moderately ill (3.9 [3.8–4.2] g/dL). Among critically ill patients, mortality was higher in those with a lower SA concentration (3.3 g/dL vs. 3.9/dL; *p* < 0.001), with an AUC of 0.79 (95% CI, 0.64–0.93; *p* < 0.001) and 3.5 g/dL as the optimal cutoff point. Similar findings also have been obtained in more recent publications and meta-analyses with different populations. In the unique comparative study with 203 healthy controls, 191 infected patients had lower SA levels in the presence of respiratory symptoms [22]. Huang et al. studied prognostic factors in 299 adults. They found that 35.5% of the patients had an SA value <3.5 g/dL, and that this variable was an independent predictor of in-hospital mortality (OR, 6.394; 95% CI, 1.315–31.092) [17]. Uyar et al. [24] also observed lower SA levels in severely ill patients (3.1 ± 0.2 g/dL vs. 4.0 ± 0.3 g/dL; *p* = 0.0001), with a poorer outcome in those with SA levels lower than 3.5 g/dL at admission (sensitivity, 76.47%; specificity, 73.81%). The authors established an SA cutoff point for admission to the ICU at <3.6 g/dL (AUC, 0.989; *p* < 0.0001; 95% CI, 0.924–1.000). Similar results were also obtained by Kheir et al. [21] in a retrospective study with 109 hospitalized patients showing that an SA lower than 3.3 g/dL was related to a higher risk of developing ARDS, ICU admission, and readmission within 90 days after discharge. Interestingly, these authors estimated a 72% decreased risk of developing venous thromboembolism for every 1 g/dL increase of SA during hospitalization. A similar cutoff value for SA was obtained by Sanson et al. in a study with 69 patients, showing that an SA lower than 3.5 g/dL was related to a higher need of any ventilatory support, PaO_2_/FiO_2_ ratio, and 60-day mortality [24]. Of interest, it has also been demonstrated that patients with acute COVID-19 initially managed at home and with an SA value < 3.5 g/dL at diagnosis have an increased risk of posterior hospital admission compared to those with a value higher than this one (15.38% vs. 4.26%, *p* = 0.06 hospitalizations); however, the multivariate analysis did not reveal differences after controlling for age, sex, and co-morbidities [23]. In general, all recent meta-analysis conclude that SA is a good predictor of evolution in hospitalized COVID-19, despite other factors and the severity [26,27,36]. As is our study, they showed that acute COVID-19 severity and survival are inversely related to SA values. In reference to the cutoff value of SA, in a meta-analysis with 23 studies, the authors established a cutoff SA value of 3.35 g/dL for severe COVID-19 [36]. According with these studies and meta-analyses, we have found that participants for whom the final composite outcome was recorded presented with significantly higher values of CRP (108.9 [96.5] mg/L vs. 70.6 [70.3] mg/L; *p* < 0.001) and lower values of SA (3.5 [0.6] g/dL vs. 3.8 [0.5] g/dL; *p* < 0.001), as well as a higher prevalence of hypoalbuminemia (42.4% vs. 21.2%; *p* < 0.001), defined as an SA <3.5 g/d. All our results are in accordance with those reported in the literature; however, it should be emphasized as a common conclusion of all the meta-analysis the high heterogeneity of the included studies [26,27,36]. As an added value, compared with previous publications, we have made an internal validation of the model.

Studies in patients with sepsis have demonstrated that some composite inflammatory parameters, such as the CRP/SA ratio, have a higher predictive capacity than CRP alone [50]. Similar observations have been made in situations other than infectious catabolic and inflammatory states, such as hospitalized older people [51], cancer patients [52], and the postoperative period [53]. As in the present study, previous studies have considered the potential capacity of the CRP/SA ratio to have greater predictive capacity than each of the components of the ratio alone in COVID-19 [28,29,30,31,32,33,34,35]. Although with a very high heterogeneity, in general, the existing literature demonstrates the prognostic capacity of CRP/SA ratio in the hospitalized COVID-19 patient, as well as that this predictive capacity is higher than that of each one of the parameters alone [36]. Some studies have also demonstrated the superiority of this ratio compared with others. More difficult than in the case of SA alone is to establish a cutoff value of this ratio mainly because of the different considered outcomes and applied statistical methods applied in each study.

In a retrospective study with 113 non-severely ill and 84 severely ill patients, Karakoyun et al. [29] estimated that the CRP/SA ratio was an independent predictive factor of severity, with severely ill patients having higher values: OR after multivariate analysis of 1.264 (*p* = 0.037), and ROC curve analysis assigning 9.0 as the cutoff value for differentiation of severe COVID-19 (AUC, 0.718; 69.1% sensitivity, 70.8% specificity; *p* < 0.001). El-Shabrawy et al. [30] found that a CRP/SA value higher than 11.4 was associated with COVID-19 mortality (hazard ratio, 26.5 [95% CI, 2.6–270.7]) after adjustment for age and comorbidities (*p* = 0.006). Torun et al. [33] found in 118 patients that severe cases had a higher value of CRP/SA compared to no severe cases (3.18 [0.16–84.12] vs. 25.62 [1.08–126.35]; *p* < 0.0001) with an AUC of about that obtained in our study (0.841). Moreover, in older population Ayranci et al. found an increased CRP/SA ratio (21.39 (6.02–55.07), 4.82 (1.17–17.03), *p* < 0.001) in patients that died during hospitalization [32]. In our study, the ratio CRP/SA has a significantly increased predictive value than each of the components separately, a finding similar to that obtained by Li et al. (AUC for CRP, SA and CRP/SA: 0.769 [95% CI: 0.709–0.829, *p* < 0.001]. These authors also demonstrated a higher predictive capacity of this ratio compared to the neutrophil-to-lymphocyte ratio, platelet-to-lymphocyte ratio, CRP/Alb ratio, and systemic immune-inflammation index [28].

In the present study, as suggested by the existing literature, a continuous and increasing relationship was found for the CRP/SA value, and prognosis was increasingly worse from values of about 10 onward. In our opinion, given the continuous direct relationship between CRP/SA and poor prognosis, it seems that 25 could be considered a reasonable operative value to determine whether COVID-19 patients have a higher risk of developing the composite outcome of in-hospital mortality, length of stay, need for admission to the ICU, and need for NIMV. This value seems to be in accordance with the values proposed by other investigators [28,29,30,32,33,34], although our suggested cutoff value of about 25 is higher than the values published elsewhere. The reasons for these differences are not clear. The main reason could be the differences in the predicted final outcome among different studies, with a cutoff value of 9 as a predictor of severity [29], 11.4 for mortality [30], and 25.0 for a composite prognostic outcome (present study). A similar to our cutoff value proposal was found in a study including 175 patients who were older than 65 years of age in which the authors established in 23 the value to predict in-hospital mortality [34]. Of interest, taking into account the use of a similar composite outcome to ours, including hospital mortality, ICU admission, invasive mechanical ventilation, and longer hospital stay, Li et al. suggest as a cutoff value for CRP/SA 18.43 as a predictor of bad evolution of severe cases [28] and as a prognostic marker of bad evolution among severe cases of COVID19. With a different approach, Güney et al. classified 275 hospitalized COVID-19 patients into tertiles depending on the value of the CRP/SA ratio at admission, with the third tertile including those participants with values between 15.9 and 111.9, finding that this group had 8.2 (95% CI: 4.2–48.1) times higher rates of in-hospital mortality compared to those with a CRP/SA lower than 2.9 [31]. In the unique validated model with a real-life independent cohort from the same center, the authors established as a cutoff for CRP/SA a value of 29.2, just taking into account the median value of the initial studied sample with 2309 participants, finding significant differences in terms of in-hospital and post-admission mortality, need for respiratory support, and some intrahospitalary complications cases [35]. Another reason for the different proposed CRP/ALB cutoff value in different publications could be the fact that, in previous publications, outcomes were qualitatively and unclearly defined. In contrast, we defined a clear composite outcome that can be objectively measured. Of interest, the study of Lucijanić M et al. [35] and ours are both the only studies that have validated the proposed model, as well as those with the higher sample size. They both proposed a quite similar CRP/SA cutoff value, 25 and 29. The methodological aspects, validations, and big sample sizes in both studies enforce the robustness of CRP/Alb as an easy prognostic indicator in hospitalized COVID-19 in terms of in-hospital complications and mortality, as well as post-admission mortality with a cutoff value indicating bad evolution of about 25 as a reasonable and operative proposal for clinical practice.

The COVID-19 pandemic should be considered a syndemic [13]. An analysis of the disease as such takes into account not only the infectious variables but also the social and nutritional variables and comorbidities, revealing biological and social interactions that are important for prognosis, treatment, and health policy [13,40,41,42]. Albumin could be a good prognostic marker because it is a global indicator of individual susceptibility, bringing together the influence of comorbidities, anti-inflammatory reactions to infection, markers of systemic inflammation [10,11,12], and nutritional status [54]. Thus, SA could explain the association between comorbidity, one of the most important determinants of the course of COVID-19, and poor prognosis [6,7]. The present study and previously published results point to the utility of SA as an easily obtainable analytical biomarker in daily clinical practice. The accuracy of this parameter can be improved by combining it with another easily obtainable parameter, CRP, to calculate the CRP/SA ratio. In a recent publication analyzing 10,238 participants from the SEMI-COVID 19 Registry (on which the present study is based) [55], the AUCs obtained for the severity of the COVID-10 multifactorial scores PSI, CURB-65, qSOFA, and MuLBSTA were 0.835, 0.825, 0.728, and 0.751, respectively. In our study, the AUC obtained for CRP/SA was similar to that obtained for the PSI and CURB-65 scores, which were the best in the abovementioned study, although recording CRP and SA at admission is somewhat easier in daily clinical practice than recording these scores.

The main limitation of the present study is its retrospective methodology, with the consequent variability in the definition of events and therapy regimens, as well as criteria for admission to the ICU and use of NIMV in each participating hospital. However, both limitations can also be considered strengths: they ensure the applicability of the conclusions because they are based on real clinical practice. The second main strength is the size of the study population, which is the largest to date (3471 patients) in studies analyzing the usefulness of SA as a prognostic marker in hospitalized COVID-19 patients. A third strength is the validation of the model.

## 5. Conclusions

Both SA and CRP values at admission are predictive indicators of in-hospital clinical course in patients admitted with COVID-19. The predictive capacity can be increased when the CRP/SA ratio is applied. Compared with multifactor prognostic scales, this parameter is easily obtainable in daily clinical practice. Although the present study is retrospective, the fact that the results are based on a large population sample and based on usual clinical practice indicates that they are robust and coherent with previous smaller studies and existing pathophysiologic knowledge about COVID-19.

Taking into account the ease of obtaining albumin and C-reactive protein values, and their short time predictive power, we believe that both parameters should be obtained at admission from all patients admitted for COVID-19, with an especial monitoring for those patients with a CRP/SA ratio value greater than 25 due to a potential poorer evolution.

## Figures and Tables

**Figure 1 biomedicines-10-01393-f001:**
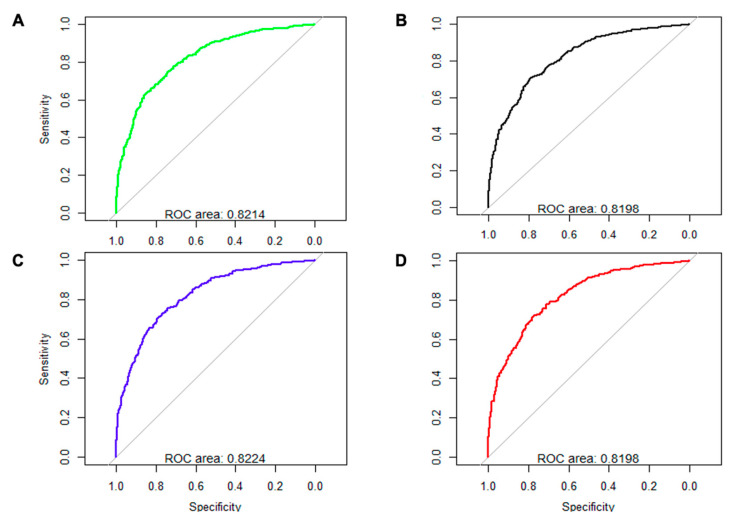
Multivariate analysis and ROC curves for baseline serum albumin alone (g/dL) (**A**), C-reactive protein (CRP) alone (mg/L) (**B**), their additive combination (**C**), and CRP/bSA ratio (**D**).

**Figure 2 biomedicines-10-01393-f002:**
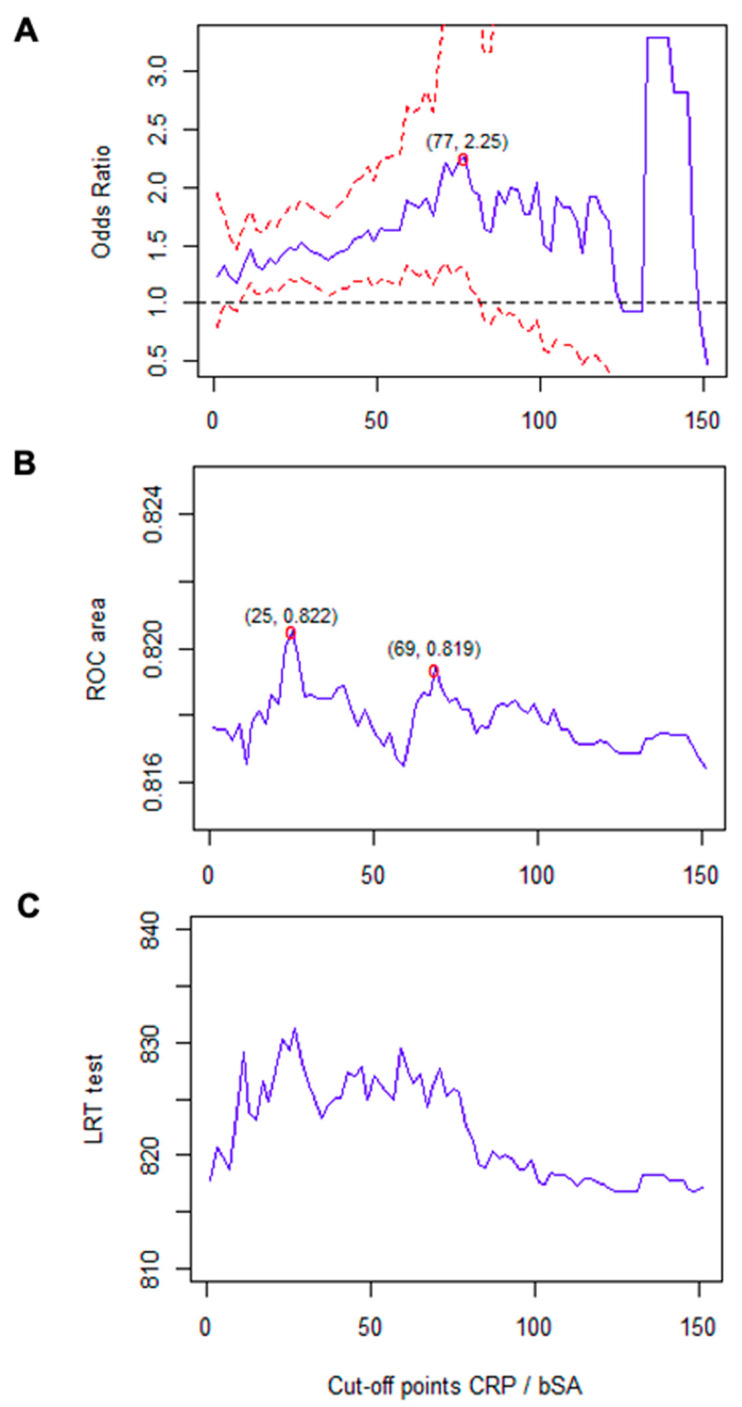
Analysis of the CRP/SA cutoff value for prediction in the multivariate model of the main outcome. (**A**) OR and 95%CI associated with the different cutoff values of CRP/bSA. (**B**) Area under the ROC curve for different cut-off values of CRP/bSA. (**C**) Analysis of goodness of fit of different cutoff values by means of the likelihood ratio test. Abbreviations: bSA, baseline serum albumin (g/dL) at admission; CRP, C-reactive protein (mg/L); LRT, likelihood ratio test.

**Table 1 biomedicines-10-01393-t001:** Baseline characteristics and pharmacological treatments of study patients.

Variable	All Patients (n = 3471)	Outcome Group (n = 1425)	Non-Outcome Group (n = 2046)	*p* *
Demographic Characteristics				
Age, years, mean (SD)	66.2 (16.0)	72.0 (14.7)	62.1 (15.6)	<0.001
Male sex, n (%)	2000 (57.6)	877 (61.5)	1123 (54.9)	<0.001
Race, n (%)				0.001
Caucasian	3100 (89.3)	1306 (91.6)	1794 (87.7)	
Latin American	322 (9.3)	100 (7.0)	222 (10.8)	
Others	49 (1.4)	19 (1.3)	30 (1.5)	
Baseline comorbidities				
CHT, n (%)	1718 (49.5)	865 (60.7)	853 (41.7)	<0.001
Dyslipidemia, n (%)	1375 (39.5)	663 (46.5)	712 (34.8)	<0.001
Smoking, n (%)				<0.001
Former smoker	897 (25.6)	437 (30.7)	460 (22.5)	
Current smoker	200 (5.8)	80 (5.6)	120 (5.8)	
Obesity, n (%)	843 (24.3)	392 (27.5)	451 (31.6)	<0.001
DM, n (%)	655 (18.9)	345 (24.2)	311 (15.2)	<0.001
Previous TIA, n (%)	265 (7.6)	63 (3.1)	89 (6.2)	<0.001
Previous IS, n (%)	152 (4.4)	54 (3.8)	37 (2.6)	<0.001
Depression, n (%)	364 (10.5)	163 (11.4)	201 (9.8)	0.127
Atrial fibrillation, n (%)	361 (10.4)	220 (15.4)	141 (6.9)	<0.001
PVD, n (%)	304 (8.8)	99 (6.9)	61 (3.0)	<0.001
Chronic anxiety, n (%)	299 (8.6)	137 (9.6)	162 (7.9)	0.091
COPD, n (%)	225 (6.5)	142 (10.0)	90 (4.4)	<0.001
Cancer, n (%)	223 (6.4)	91 (6.4)	93 (4.5)	0.017
Asthma, n (%)	199 (5.7)	95 (6.7)	170 (8.3)	0.073
CTD, n (%)	188 (5.4)	51 (3.6)	53 (2.6)	0.093
Neurodeg. disease, n (%)	184 (5.3)	172 (12.0)	99 (4.8)	<0.001
Alcohol abuse, n (%)	175 (5.0)	89 (6.2)	86 (4.2)	0.007
Angina, n (%)	166 (4.8)	53 (3.7)	44 (2.2)	0.006
AMI, n (%)	271 (7.8)	96 (6.7)	70 (3.4)	<0.001
CLD, n (%)	160 (4.6)	35 (2.4)	45 (2.2)	0.620
CHF, n (%)	97 (2.8)	146 (10.2)	79 (3.9)	<0.001
OSAS, n (%)	89 (2.6)	122 (8.6)	109 (5.3)	<0.001
CKD, n (%)	80 (2.3)	144 (10.1)	79 (3.9)	<0.001
Dependence status, n (%)				<0.001
Absent/mild	2980 (85.9)	1104 (77.5)	1876 (91.7)	
Moderate	296 (8.5)	191 (13.4)	105 (5.1)	
Severe	195 (5.6)	130 (9.1)	65 (3.2)	
CCI, points, mean (SD)	3.4 (2.6)	4.4 (2.7)	2.7 (2.3)	<0.001
CCI ≥ 3, n (%)	2885 (57.6)	1590 (55.1)	1295 (44.9)	<0.001
Baseline pharmacological treatments				
Immunosuppressants, n (%)				
Systemic corticosteroids	156 (4.5)	101 (7.1)	55 (2.7)	<0.001
Inhaled corticosteroids	337 (9.7)	175 (12.3)	162 (7.9)	<0.001
Monoclonal antibodies	34 (1.0)	15 (1.1)	19 (0.9)	0.715
Cardiovascular agents, n (%)				
ACEi	567 (16.3)	284 (19.9)	283 (13.8)	<0.001
ARB	652 (18.8)	327 (22.9)	325 (15.9)	<0.001
Statins	1131 (32.6)	542 (38.0)	589 (28.8)	<0.001
Aspirin	477 (13.7)	253 (17.7)	224 (10.9)	<0.001
Acenocoumarol	185 (5.3)	125 (8.8)	60 (2.9)	<0.001
Direct anticoagulants	149 (4.3)	79 (5.5)	70 (3.4)	
LMWH	23 (0.7)	18 (1.3)	5 (0.2)	
Antidiabetic agents, n (%)				
Insulin	202 (5.8)	115 (8.1)	87 (4.3)	<0.001
Metformin	455 (13.1)	227 (15.9)	228 (11.1)	<0.001
SGLT2i	78 (2.2)	30 (2.1)	48 (2.3)	0.638
DPP4i	242 (7.0)	125 (8.8)	117 (5.7)	0.001

ACEi, angiotensin-converting enzyme inhibitors; AMI, acute myocardial infarction; ARB, angiotensin receptor blocker; CCI, age-adjusted Charlson Comorbidity Index; CHF, chronic heart failure; CHT, chronic hypertension; CKD, chronic kidney disease; CLD, chronic liver disease; COPD, chronic obstructive pulmonary disease; CTD, connective tissue disease; DM, diabetes mellitus; DPP4i, dipeptidyl peptidase-4 inhibitors; IS, ischemic stroke; LMWH, low-molecular-weight heparin; Neurodeg., neurodegenerative disease; OSAS, obstructive sleep apnea syndrome; PVD, peripheral vascular disease; SGLT2i, sodium–glucose transporter (SGLT) 2 inhibitors; TIA, transient ischemic attack; Med, median. * Comparison of patients who reached and did not reach the final composite endpoint.

**Table 2 biomedicines-10-01393-t002:** COVID-19-related clinical manifestations and signs of study patients.

Variable	All Patients (n = 3471)	Outcome Group (n = 1425)	Non-Outcome Group (n = 2046)	*p* *
Hospitalization				
Origin of infection, n (%)				<0.001
Community	2983 (85.9)	1151 (80.8)	1832 (89.5)	
Healthcare staff	245 (7.1)	142 (10.0)	103 (5.0)	
Others	243 (7.0)	132 (9.3)	111 (5.4)	
Duration of symptoms before admission, days, mean (SD)	6.9 (4.4)	6.2 (4.5)	7.5 (4.3)	<0.001
Clinical symptoms				
Cough, n (%)				<0.001
Absent	849 (24.5)	396 (27.8)	453 (22.1)	
Dry	2072 (59.7)	781 (54.8)	1291 (63.1)	
Wet	550 (15.8)	248 (17.4)	302 (14.8)	
Dyspnea, n (%)	2021 (58.2)	949 (66.6)	1072 (52.4)	<0.001
Asthenia, n (%)	1499 (43.2)	561 (39.4)	938 (45.8)	<0.001
Myalgia, n (%)	1153 (33.2)	375 (26.3)	778 (38.0)	<0.001
Diarrhea, n (%)	878 (25.3)	289 (20.3)	589 (28.8)	<0.001
Anorexia, n (%)	634 (18.3)	282 (19.8)	352 (17.2)	0.058
Nausea, n (%)	438 (12.6)	158 (11.1)	280 (13.7)	0.027
Headache, n (%)	414 (11.9)	138 (9.7)	276 (13.5)	0.001
Odynophagia, n (%)	318 (9.2)	110 (7.7)	208 (10.2)	0.014
Ageusia, n (%)	276 (8.0)	49 (3.4)	227 (11.1)	<0.001
Vomiting, n (%)	263 (7.6)	107 (7.5)	156 (7.6)	0.951
Anosmia, n (%)	240 (6.9)	44 (3.1)	196 (9.6)	<0.001
Abdominal pain, n (%)	215 (6.2)	84 (5.9)	131 (6.4)	0.590
Physical examination				
Confusion, n (%)	334 (9.6)	244 (17.1)	90 (4.4)	<0.001
Tachypnea, n (%)	1128 (32.5)	669 (46.9)	459 (22.4)	<0.001
SBP, mmHg, mean (SD)	128.7 (20.5)	128.6 (21.3)	128.8 (19.9)	0.815
DBP, mmHg, mean (SD)	74.4 (12.8)	72.7 (12.9)	75.6 (12.7)	<0.001
Heart rate, bpm, mean (SD)	88.9 (17.2)	89.0 (16.8)	88.8 (17.7)	0.736
Temperature, °C, mean (SD)	37.1 (1.0)	37.0 (0.9)	37.3 (1.0)	<0.001
Temperature ≥ 38 ° C, n (%)	2238 (64.5)	910 (63.9)	1328 (64.9)	0.019
Crackles, n (%)	1806 (52.0)	812 (57.0)	994 (48.6)	<0.001
Wheezing, n (%)	223 (11.7)	124 (8.7)	99 (4.8)	<0.001
Rhonchi, n (%)	406 (11.7)	232 (16.3)	174 (8.5)	<0.001
Baseline O_2_ saturation, %, mean (SD)	93.3 (5.5)	94.9 (3.4)	90.9 (6.9)	<0.001
Baseline O_2_ saturation <93%, n (%)	1598 (31.9)	829 (58.2)	554 (27.1)	<0.001
Thoracic radiological findings				
Alveolar condensation, n (%)				<0.001
Absent	1782 (51.3)	683 (47.9)	1099 (53.7)	
Unilateral	586 (16.9)	215 (15.1)	371 (18.1)	
Bilateral	1103 (31.8)	527 (37.0)	576 (28.2)	
Interstitial infiltrate, n (%)				<0.001
Absent	1208 (34.8)	497 (34.9)	711 (34.8)	
Unilateral	367 (10.6)	111 (7.8)	256 (12.5)	
Bilateral	1896 (54.6)	817 (57.3)	1079 (52.7)	
Pleural effusion, n (%)				0.001
Absent	3344 (96.3)	1352 (94.9)	1992 (97.4)	
Unilateral	87 (2.5)	50 (3.5)	37 (1.8)	
Bilateral	40 (1.2)	23 (1.6)	17 (0.8)	
Analytical findings				
Glucose, mg/dL, mean (SD)	124.8 (49.9)	136.7 (59.4)	116.6 (40.1)	<0.001
Creatinine, mg/dL, mean (SD)	1.1 (0.83)	1.28 (1.0)	0.95 (0.62)	<0.001
Sodium, mEq/L, mean (SD)	137.6 (4.4)	137.6 (5.1)	137.6 (3.8)	0.922
Potassium, mEq/L, mean (SD)	4.1 (0.5)	4.2 (0.6)	4.1 (0.5)	<0.001
Hemoglobin, g/dL, mean (SD)	13.8 (1.9)	13.4 (2.0)	14.0 (1.7)	<0.001
Leukocytes’ 10^6^/L, mean (SD)	7074.0 (4668.6)	7629.7 (4702.5)	6687.0 (4606.5)	<0.001
Lymphocytes’ 10^6^/L, mean (SD)	1080.5 (888.6)	1001.6 (1119.2)	1135.5 (678.4)	<0.001
Neutrophils’ 10^6^/L, mean (SD)	5216.4 (3031.6)	593 (3.46)	400.7 (2.6)	<0.001
Platelets’ 10^9^/L, mean (SD)	20,5182.7 (85,858.2)	200.0 (91.6)	208.7 (81.5)	0.004
CRP, mg/L, mean (SD)	86.3 (84.2)	108.9 (96.5)	70.6 (70.3)	<0.001
Albumin, g/dL, mean (SD)	3.7 (0.5)	3.5 (0.6)	3.8 (0.5)	<0.001
CRP/SA, mean (SD)	24.4 (25.1)	31.9 (29.2)	19.2 (20.2)	<0.001

SBP, systolic blood pressure; DBP, diastolic blood pressure; bpm, beats per minute; CRP, C-reactive protein; SA, serum albumin; SD, Standard deviation. * Comparison of patients who reached the final outcome vs. those who did not.

**Table 3 biomedicines-10-01393-t003:** Multivariate logistic regression model for the additive form (SA + CRP).

Variable	Odds Ratio (95% CI) *	*p*
Age	1.015 (1.004–1.026)	0.008
Baseline SA	0.674 (0.551–0.826)	<0.001
CRP	1.002 (1.001–1.004)	0.003
Community origin	2.469 (1.654–3.685)	<0.001
Dyspnea	1.273 (1.034–1.568)	0.023
Confusion	1.843 (1.278–2.659)	0.001
Tachypnea	1.591 (1.272–1.991)	<0.001
Baseline systemic corticosteroids	2.288 (1.441–3.632)	<0.001
Days with symptoms	0.939 (0.917–0.961)	<0.001
Age-adjusted Charlson score	1.100 (1.029–1.175)	0.005
Temperature	1.279 (1.152–1.420)	<0.001
Baseline 0_2_ saturation	0.889 (0.866–0.912)	<0.001
Platelet count	0.997 (0.996–0.998)	<0.001

SA, serum albumin; CRP, C-reactive protein. * Odds ratio adjusted for race, radiological findings, main clinical manifestations (ageusia, dyspnea, confusion, and tachypnea), diabetes mellitus, heart rate, platelet and neutrophil count, ALT, hemoglobin, and fasting glucose values.

## Data Availability

Data from the study are available in a specific database owned by Sociedad Española de Medicina Interna (https://covid.reginus.es, accessed on 3 June 2022).

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
