# Peer review of "C-Reactive Protein and Serum Albumin Ratio: A Feasible Prognostic Marker in Hospitalized Patients with COVID-19"

_biomedicines, 2022, doi:10.3390/biomedicines10061393_

Round 1
Reviewer 1 Report
In the study “C-reactive protein and serum albumin ratio: a feasible prognostic marker in hospitalized patients with COVID-19” Vicente Giner-Galvan and colleagues have demonstrated the predictive capacity of serum albumin and C-reactive protein combination as a prognostic marker in covid-19 patients.
Though is a retrospective work, it is adding some new insights into the ongoing covid-19 prognosis research.
The study is comprising a very big sample size which is a strong point for the validation of the data.
Overall it is a good piece of work.
There are some minor comments those need to be addressed.
1- Please spell-check the title of the paper.
2- Why not the patients under the age of 18 were included? (Line 103)
3- It should be real-time reverse transcription PCR. (Line 105)
4- What could be the effect of vaccination against covid-19 on the level of SA and CRP? And will the cut-off value of 25 (suggested by the authors) will get influenced by the vaccination.
Author Response
- Please spell-check the title of the paper.
We have changed the title of the manuscript. We have changed the initial title “C-reactive protein and serum albumin ratio: a feasible prognotic marker in hospitalized patients with COVID-19” to “C-reactive protein and serum albumin ratio: a feasible prognostic marker in hospitalized patients with COVID-19”.
- Why not the patients under the age of 18 were included? (Line 103).
Because all the investigators were working in different Spanish General Internal Medicine departments, in which all the patients are adults. In Spain people under 18 years are usually attended by pediatricians. On the other hand it is extensively known the different profile of COVID on pediatric populations compared with adults.
- It should be real-time reverse transcription PCR. (Line 105).
According to this suggestion, we have changed the text of the initial manuscript “… reverse transcription polymerase chain reaction (RT-PCR) testing…” to “… real-time reverse transcription polymerase chain reaction (RT-PCR) testing…” as suggested by the reviewer.
- What could be the effect of vaccination against covid-19 on the level of SA and CRP? And will the cut-off value of 25 (suggested by the authors) will get influenced by the vaccination.
All the included people were those infected during the two first epidemic waves in Spain. At this time we did not have any vaccine. Because of this reason it is absolutely speculative any comment about the effect of vaccination in the present study and SA and CRP values. We can speculate a confounder effect due to an increased inflammation as a reaction to the vaccine. It is another possibility a smoother relationship because the vaccination is related to mild forms of acute infection. Finally, it is also possible no effect on the predictive value of the PCR/albumin ratio. For these reasons it is very difficult to establish a different cut off value considering the vaccination situation, although it could be a very interesting issue to be undertaken probably comparing a group of vaccinated people against another one without previous vaccination, although in Spain, because of a very high percentage of population vaccination, it would be very difficult to obtain non vaccinated people.
Reviewer 2 Report
The scientific studies that investigate inflammatory markers or predictorsof pathologies or health conditions are now numerous.
This work, however, adds information of considerable scientific importance to
what we have been trying to study for years regarding Covid 19 and its effects.
Many years from now, the information that comes out of surveys like these
will be helpful in understanding further investigations.
Author Response
This work, however, adds information of considerable scientific importance to what we have been trying to study for years regarding Covid 19 and its effects.
Thank you for your kind comments, especially when you indicate that our study “adds information of considerable scientific importance”, because we understand that the only reason for a scientific publication is to improve the state of the knowledge at the time of its publication.
Reviewer 3 Report
The work entitled " C-reactive protein and serum albumin ratio: a feasible prognotic marker in hospitalized patients with COVID-19" aims to analyze the prognostic capacity of baseline serum abumin (SA) and CRP for a composite outcome comprising mortality, length of stay, ICU admission, and non-invasive mechanical ventilation in hospitalized COVID-19 patients. This is a pertinent paper. The paper is well written and has the makings of a publication. However, authors could provide some tips for implementation based on presented data. How could be scored against a validation cohort? An algorithm should be proposed.
Some typo mistakes must be corrected, in title “prognotic” should be “prognostic”
Author Response
1.- This is a pertinent paper. The paper is well written and has the makings of a publication.
Thank you for your kind general comment.
2.- Authors could provide some tips for implementation based on presented data.
According to the revision´s suggestions we have added in the conclusions the phrase “Taking into account the ease of obtaining albumin and C-reactive protein values at admission, and their predictive power, we believe that both parameters should be obtained from all patients admitted for COVID-19, and monitoring should be implemented for those with an CRP/SA ratio value greater than 25.”
3.- How could be scored against a validation cohort?.The only way to validate the model in a definitive way should be making a prospective study comparing patients depending on the values of SA, CRP and its quotient. But it would be a different study. However, as stated in the “statistical analysis” section of the manuscript, an indirect validation analysis was made. In it, as stated in lines 161 to 165, “The optimal predictive cutoff point for CRP/SA was calculated. After a final validation of the model obtained, the total sample was randomly divided into a training group (70%) to build the model and a testing group (30%) to validate it. Validation consisted of using the model adjusted in the training group with the testing group data and calculating the probability of occurrence of the final outcome.” 4.-An algorithm should be proposed. In our opinion, the final statement in the “conclusions” section establishing the value of CRP/SA to be considered as a short-time prognostic value and clearly indicating that it should be measured at admission for each hospitalized COVID-19 patient, is a way to propose the algorithm suggested by the reviewer.
5.- Some typo mistakes must be corrected, in title “prognotic” should be “prognostic”.
We have changed the title of the manuscript. We have changed the initial title “C-reactive protein and serum albumin ratio: a feasible prognotic marker in hospitalized patients with COVID-19” to “C-reactive protein and serum albumin ratio: a feasible prognostic marker in hospitalized patients with COVID-19”.
Reviewer 4 Report
First of all the article has a list of authors but we could not understand who are the authors of this article?
Secondly the article is not structured following the indications of the journal.
The abstract has 241 words, when the maximum is 200.
What brings this article new from the one published in 2020?
Regarding the discussions i would suggest the authors to include more recent published articles.
Author Response
1.- First of all the article has a list of authors but we could not understand who are the authors of this article?.
As established in the Instructions for authors, we have specified the task that each of the signatories of the paper has had. Specifically, in the “Authors contribution” it is specified:
“Conceptualization, Vicente Giner-Galvañ, Francisco Pomares-Gómez; Data curation, Vicente Giner-Galvañ, José Quesada, Javier Tejada-Montes, Jesús Baltasar-Corral, María Taboada-Martínez, Blanca Sánchez-Mesa, Francisco Arnalich-Fernández, Esther del Corral-Beamonte, Almudena López-Sampalo, Paula Pesquera-Fontán, Mar Fernández-Garcés, Ricardo Gómez-Huelgas and José Ramos-Rincón; Formal analysis, Francisco Pomares-Gómez and José Quesada; Funding acquisition, Vicente Giner-Galvañ; Investigation, Vicente Giner-Galvañ, Manuel Rubio-Rivas, Javier Tejada-Montes, Jesús Baltasar-Corral, María Taboada-Martínez, Blanca Sánchez-Mesa, Francisco Arnalich-Fernández, Esther del Corral-Beamonte, Almudena López-Sampalo, Paula Pesquera-Fontán, Mar Fernández-Garcés, Ricardo Gómez-Huelgas and José Ramos-Rincón; Methodology, Vicente Giner-Galvañ, Francisco Pomares-Gómez, Ricardo Gómez-Huelgas and José Ramos-Rincón; Project administration, Vicente Giner-Galvañ, Ricardo Gómez-Huelgas and José Ramos-Rincón; Supervision, Vicente Giner-Galvañ and Francisco Pomares-Gómez; Validation, Vicente Giner-Galvañ and Francisco Pomares-Gómez; Writing – original draft, Vicente Giner-Galvañ; Writing – review & editing, Vicente Giner-Galvañ, Francisco Pomares-Gómez, Manuel Rubio-Rivas, Javier Tejada-Montes, Jesús Baltasar-Corral, María Taboada-Martínez, Blanca Sánchez-Mesa, Francisco Arnalich-Fernández, Esther del Corral-Beamonte, Almudena López-Sampalo, Paula Pesquera-Fontán, Mar Fernández-Garcés, Ricardo Gómez-Huelgas and José Ramos-Rincón.”
As the correspondence author, V Giner-Galvañ is the reference author with the main contributions to the article, although all the rest of the authors are also authors according to international criteria. They all have had a more direct participation in the actual paper than the rest of the members in the Spanish national Register SEMI-COVID.
2.- Secondly the article is not structured following the indications of the journal.
We have reviewed the structure of the actual manuscript version comparing with that established in the instructions for the authors for “Research manuscript” in Biomedicines. After comparation, we do not understand what structural aspects of the manuscript need to be modified. We will be very grateful to this reviewer if he/she indicates us exactly which changes have to be made. As an indication that we have applied the original indications of Biomedicines, we have applied the specific template prepared by the publication for this kind of manuscripts.
3.- The abstract has 241 words, when the maximum is 200.
We have again applied the counting tool from Word. We have adjusted the abstract to the maximum length (200 words), although some differences in the word count can be present depending probably on the version of the electronic tool. At the actual version we have applied the function of word for word count, and the actual count is 196 including the headlines of each part of the abstract.
4.- What brings this article new from the one published in 2020?.
We do not understand exactly which article the reviewer is referring to due to the high number of different papers published during the mentioned year. We could answer if the exact article is indicated to us.
We are not sure, but it is probable that the reviewer is referring to a previous study in Spanish population, that one of De la Rica et al. (De la Rica R, Borges M, Aranda M, del Castillo A, Socias A, Payeras A, et al. Low albumin levels are associated with poorer outcomes in a case series of COVID-19 patients in Spain: A retrospective cohort study. Microorganisms. 2020; 8 (8): 1106). If it is the article, we think there is a lot of differences compared with ours. The main is the different analyzed population: in the study by De la Rica patients that had required ICU admission were compared to those that did not need such admission. But the main difference from our point of view is related to the fact that our study has a much bigger sample, as well as the multicentric nature of ours, assuring a sufficient representativity of the Spanish population and usual clinical practice in our country.
In a general way we think that our study compared with previous ones brings some aspects on the potential of albumin as a predictor in COVID-19 hospitalized patients. The first one is the sample size, the highest considering all previous studies. The second aspect is the fact that we have made an inner validation of the model, with just one previously published study making this validation. The used methodology to suggest an operative CRP/SA cut-off value as well as the congruence with the previously published works is another aspect to be considered that, in our opinion, reinforces the predicting value of SA in hospitalized COVID-19. Finally, the national multicentric nature of our study and the fact that data have been obtained from the usual clinical practice, allows to assume a high representativity of the findings, with the multicentric nature of the study as a unique characteristic of our study compared with previous ones.
5.- Regarding the discussions i would suggest the authors to include more recent published articles.
It is quite difficult to actualize the bibliography in a field as COVID, in which there is a constant flux of publications. On the other hand, the submission process itself make difficult a continuous actualization. We have reviewed again all the published bibliography. We have included in the new version of the initially submitted manuscript those in our opinion significant new publications in the Introduction and Discussion sections.
After the review of the recent bibliography we considered that it has not been significant differences in the field, with a lot of small sample studies that globally do not affect significantly the content and findings of our work. It has to be underscored the publication of some interesting recent meta-analysis such that of Paliogiannis (Paliogiannis P, Mangoni AA, Cangemi M, Fois AG, Carru C, Zinellu A. Serum albumin concentrations are associated with disease severity and outcomes in coronavirus 19 disease (COVID-19): a systematic review and meta-analysis. Clin Exp Med. 2021; 21 (3): 343-54) and Qin (Qin R, He L, Yang Z, Jia N, Chen R, Xie J, Fu W, Chen H, Lin X, Huang R, Luo T, Liu Y, Yao S, Jiang M, Li J. Identification of Parameters Representative of Immune Dysfunction in Patients with Severe and Fatal COVID-19 Infection: a Systematic Review and Meta-analysis. Clin Rev Allergy Immunol. 2022: 1-33) as a tool to obtain a global idea of the field. Theses meta-analysis reinforce the aforementioned improvement in the field that our work can contribute as its multicentric nature, the higher sample size and the validation of the proposed model.
Apart from the mentioned meta-analysis, we think that the probably more striking study is that of Lucijanić et al (Lucijanić M, Stojić J, Atić A, Čikara T, Osmani B, Barišić-Jaman M, Andrilović A, Bistrović P, Zrilić Vrkljan A, Lagančić M, Milošević M, Vukoja I, Đerek L, Lucijanić T, Piskač Živković N. Clinical and prognostic significance of C-reactive protein to albumin ratio in hospitalized coronavirus disease 2019 (COVID-19) patients: Data on 2309 patients from a tertiary center and validation in an independent cohort. Wien Klin Wochenschr. 2022; 134 (9-10): 377-84. doi: 10.1007/s00508-021-01999-5), a study with a sample size near to us and a validation of its model although, as an important difference compared with our study, its unicentric nature as a limitation for the generalization of its findings.
Round 2
Reviewer 3 Report
The paper is acceptable for publication.
Reviewer 4 Report
The paper is suitable for publication.